# Optimal PD-L1–high cutoff for association with overall survival in patients with urothelial cancer treated with durvalumab monotherapy

**Magdalena Zajac[1]☯, Jiabu Ye[2]☯, Pralay Mukhopadhyay[2], Xiaoping Jin[3¤a], Yong Ben[4¤b], Joyce Antal[5¤c], Ashok K. Gupta[6], Marlon C. Rebelatto[6], J. Andrew Williams[1], Jill Walker[1]***

**1** AstraZeneca, Cambridge, England, United Kingdom, **2** AstraZeneca, Gaithersburg, Maryland, United States of America, **3** MedImmune, Gaithersburg, Maryland, United States of America, **4** AstraZeneca, Gaithersburg, Maryland, United States of America, **5** MedImmune, Gaithersburg, Maryland, United States of America, **6** MedImmune, Gaithersburg, Maryland, United States of America

☯ These authors contributed equally to this work.
¤a Current address: Akesobio Guangdong Province, China
¤b Current address: BeiGene, San Francisco, California, United States of America
¤c Current address: G1 Therapeutics Inc, Research Triangle Park, North Carolina, United States of America
* Jill.Walker@astrazeneca.com

**Data Availability Statement:** This has been updated below to include the the web link to a request form Data underlying the findings

## Abstract

### Background

Studies have indicated that programmed death ligand 1 (PD-L1) expression may have utility as a predictive biomarker in patients with advanced/metastatic urothelial carcinoma (UC). Different immunohistochemical (IHC) assays are in development to assess PD-L1 expression on tumor cells (TCs) and tumor-infiltrating immune cells (ICs).

### Methods

In this *post hoc* analysis of the single-arm, phase 1/2 Study 1108 (NCT01693562), PD-L1 expression was evaluated from tumor samples obtained prior to second-line treatment with durvalumab in patients with advanced/metastatic UC using the VENTANA (SP263) IHC Assay. The primary objective was to determine whether the TC $\geq$ 25%/IC $\geq$ 25% algorithm (i.e., cutoff of $\geq$ 25% TC or $\geq$ 25% IC with PD-L1 staining at any intensity above background) was optimal for predicting response to durvalumab. PD-L1 expression data were available from 188 patients.

### Results

After a median follow-up of 15.8 and 14.6 months, higher PD-L1 expression was associated with longer overall survival (OS) and progression-free survival (PFS), respectively, with significant separation in survival curves for PD-L1–high and–low expressing patients for the TC $\geq$ 25%/IC $\geq$ 25% cutoff (median OS: 19.8 vs 4.8 months; hazard ratio: 0.46; 90% confidence interval: 0.33, 0.639). OS was also prolonged for PD-L1–high compared with–low

described in this manuscript may be obtained in accordance with AstraZeneca's data sharing policy described at https://astrazenecagrouptrials. pharmacm.com/ST/Submission/Disclosure Anonymized datasets may be available on request. Requests for access to data may be submitted at https://astrazenecagroup-dt.pharmacm.com/DT/ Home. The request will undergo an internal review process, and if approved, data will be prepared and shared with specified accessors named on the request form for 12 months.

**Funding:** The funder (AstraZeneca) provided support in the form of salaries for all authors, who contributed to the study design, data collection and analysis, decision to publish, or preparation of the manuscript. The specific roles of these authors are articulated in the 'author contributions' section.

**Competing interests:** J. Ye, P. Mukhopadhyay, J. Walker, A. K. Gupta, M. C. Rebelatto, J. A. Williams are employees of AstraZeneca. M. Zajac, Y. Ben, X. Jin, J. Antal were employees of MedImmune/ AstraZeneca at the time of manuscript development. These commercial affiliations do not alter our adherence to PLOS ONE policies on sharing data and materials.

patients when samples were categorized using TC/IC combined positive score $\geq$ 10 and IC$\geq$ 5% cutoffs. In multivariate analysis, IC but not TC PD-L1 expression was significantly associated with OS, PFS, and objective response rate ($P$ < 0.001 for each), although interaction analysis showed similar directionality of benefit for ICs and TCs.

## Conclusions

These findings support the utility of a combined TC/IC algorithm for predicting response to durvalumab in patients with UC, with the TC$\geq$ 25%/IC$\geq$ 25% cutoff optimal when used with the VENTANA (SP263) IHC Assay.

## Introduction

The programmed cell death-1 receptor (PD-1) and ligand (PD-L1) pathway is an important checkpoint for immune tolerance in normal physiology, but also plays a role in immune escape in cancer [1,2]. PD-L1 expression is often associated with tumor cells (TCs), but PD-L1– expressing tumor-infiltrating immune cells (ICs) may also contribute to the dynamic microenvironment of the tumor and host [3]. The clinical utility of PD-L1 expression for predicting response to PD-1/PD-L1 directed immunotherapies has been investigated in a variety of tumors, including urothelial carcinoma (UC) [4,5]. A number of studies investigating the antitumor effects of checkpoint blockade using anti–PD-1/PD-L1 immuno-oncology (IO) agents showed higher objective response rates (ORR) in patients with advanced/metastatic UC with TC and IC PD-L1 expression above given cutoffs compared with PD-L1 expression below given cutoffs [6–10].

The US Food and Drug Administration (FDA) and European Medicines Agency (EMA) have recently cautioned against the use of single-agent checkpoint inhibition for the treatment of PD-L1–low UC in first line cisplatin-ineligible patients [11]. Therefore, the determination of PD-L1 expression before IO treatment provides an opportunity to optimize the selection of patients who are most likely to respond to therapy. Currently, there are four commercial PD-L1 immunohistochemical (IHC) diagnostic assays approved by the FDA to evaluate PD-L1 as a biomarker in different tumors, including UC [12–14]. These assays utilize different antibodies, cutoffs, and algorithms for classifying samples as PD-L1 high or low/negative; for ICs, IHC analysis may involve cytoplasmic as well as membrane staining. The scoring algorithms, assay-specific monoclonal antibody, and visualization reagents may contribute to diverse assay specificity and sensitivity performance [15]. The VENTANA (SP263) Assay algorithm is used in the context of UC tissue samples to measure PD-L1 expression as the percentage of TC or IC staining at any intensity above background. The latter is further defined as the percentage of the IC area within the tumor exhibiting PD-L1 IC staining [16]. The cutoff for PD-L1–high expression of $\geq$ 25% in both compartments was chosen since it appeared to enrich for ORR in data from the first 20 patients with UC who received durvalumab [7]. Other assays use different algorithms and cutoffs. The algorithm for the PD-L1 IHC 22C3 pharmDx assay is used to classify UC samples wherein PD-L1 expression is measured as the number of PD-L1–stained TCs and ICs relative to the total number of all TCs (giving a combined positive score [CPS]), with a cutoff of $\geq$ 10 used for positive PD-L1 expression in one study of patients with UC [17,18]. PD-L1 expression in the context of UC samples for the VENTANA (SP142) IHC Assay algorithm is measured as the proportion of tumor area occupied by PD-L1– expressing IC at any staining intensity (IC$_{TCArea}$), with a cutoff $\geq$ 5% for PD-L1–high

expression [19]. For the PD-L1 IHC 28–8 pharmDx assay, PD-L1 expression is also measured as the percentage of TCs exhibiting membrane staining at any intensity, with a protocol amendment in a study of patients with UC used to change the definition of PD-L1 positive from $\geq$ 1% to a cutoff of $\geq$ 5% [20,21].

Evidence supporting the immunogenic effects of bacillus Calmette-Guérin in UC points to a role for immune stimulation in this disease and provides a rationale for targeting PD-L1 expressed on ICs [22,23]. PD-L1 expression on TCs has been correlated with clinical response to IO agents, with the upregulation of PD-L1 supporting the hypothesis that PD-L1 is an important mechanism for immune escape through its role as an inhibitory ligand [24]. However, the relative importance of TC and IC PD-L1 expression in predicting response to IO therapy remains unclear and may vary across cancer types.

Durvalumab is a selective, high-affinity, Fc-engineered human IgG1 antibody that blocks PD-L1 binding to PD-1 and CD80. Durvalumab was approved by the US FDA for patients with locally advanced/metastatic UC who have disease progression during or following platinum-containing chemotherapy. Durvalumab was approved based on the results of an interim analysis of the UC cohort from a single-arm, phase 1/2 open-label study in patients with advanced solid tumors (Study 1108) [25,26]. Although initial analysis has previously validated the TC $\geq$ 25%/IC $\geq$ 25% cutoff for PD-L1 expression in UC [7], there is an opportunity to further confirm this cutoff as optimal in a larger patient dataset from 1108 study and for multiple endpoints.

The findings of a *post hoc* analysis of the cohort of patients with locally advanced/metastatic UC enrolled in Study 1108 and treated with durvalumab as second-line or subsequent therapy is presented. The *post hoc* analysis used the VENTANA PD-L1 (SP263) IHC Assay, the only PD-L1 assay validated to assess the treatment benefit of durvalumab, to evaluate the relationship between TC and IC PD-L1 expression and multiple outcomes including survival [16]. The objective was to confirm whether the TC $\geq$ 25%/IC $\geq$ 25% cutoff remained optimal for predicting response to durvalumab using a larger patient dataset than previously studied, and to compare it with other emerging scoring algorithms.

## Materials and methods

### Study design and patients

The study design, inclusion and exclusion criteria, patient population, demographics, and clinical outcomes of the UC cohort enrolled in the single-arm, phase 1/2 1108 (NCT01693562) study have been reported elsewhere [25]. Study recruitment took place from August 2014 through June 2016. Data cutoff for the post hoc analysis reported here was October 2017. In this analysis, PD-L1 expression was evaluated retrospectively using tumor tissue samples obtained prior to treatment with durvalumab monotherapy and based on methods described elsewhere [7].

Study 1108 was conducted according to the Declaration of Helsinki and approved by the independent ethics committee or institutional review board at each of the 77 participating centers (full names of the ethics committees can be found in the supporting information, S1), with written informed consent obtained from all patients; all data were anonymized prior to analysis. The inclusion of patients' data in this post hoc analysis was allowed under the Study 1108 consent form. Study participants were not compensated. This study is registered with ClinicalTrials.gov, number NCT01693562 and EudraCT number 2012-002206-52.

### Procedures and PD-L1 cutoff selection

A central laboratory assessed pretreatment tumor samples for PD-L1 expression using the VENTANA PD-L1 (SP263) IHC assay (Ventana Medical Systems, Oro Valley, AZ) optimized

for use on the automated BenchMark ULTRA platform (Ventana) [16]. The first 20 patients in Study 1108 were enrolled irrespective of PD-L1 expression.

For the purposes of the present analysis, PD-L1 expression was categorized as high or low based on PD-L1 TC and IC expression above predefined cutoffs of $\geq 1\%$, $\geq 10\%$, $\geq 25\%$, and $\geq 50\%$, and for combined TC/IC expression of $\geq 1\%/\geq 1\%$, $\geq 10\%/\geq 25\%$, $\geq 25\%/\geq 25\%$, and $\geq 50\%/\geq 25\%$. PD-L1–high TC expression was defined as the percentage of TC with membrane staining at any intensity above the predefined cutoffs. PD-L1–high IC expression was defined as the percentage of IC area present in the tumor sample exhibiting PD-L1 staining above the predefined cutoffs [16]. Conversely, the criterion for PD-L1–low expression was met if any sample did not reach the required TC, IC, or TC/IC cutoffs for high expression. PD-L1 expression was also categorized as high or low based on CPS $\geq 10$ and $IC_{TCArea} \geq 5\%$ (IC $\geq 5\%$) cutoffs. CPS $\geq 10$ and IC $\geq 5\%$ cutoffs have been applied to other assays and considered optimal in clinical trials of agents that target the PD-1/PD-L1 pathway [27]. To assess the clinical utility of these cutoffs, overall survival (OS) in patients from Study 1108 was also assessed based on PD-L1 expression above and below these cutoffs.

## Statistical analyses

To evaluate any association of PD-L1 expression with efficacy, cox regression models were used to analyze the impact of TC and IC PD-L1 expression on OS and progression-free survival (PFS). For a given choice of cutoff, cases were classified into PD-L1 positive and PD-L1 negative, and logistic regression analysis was performed to evaluate the impact of TC and IC PD-L1 expression on ORR and tumor shrinkage 15 months after the last subject was randomized. Linear regression was used to evaluate best percentage tumor change. Precision of the estimate and uncertainty in the chosen cutoff value was assessed using a two-fold cross-validation method. Kaplan–Meier plots were generated to explore the impact of single biomarker and combined TC $\geq 25\%$/IC $\geq 25\%$, CPS $\geq 10$, and IC $\geq 5\%$ algorithms on OS and PFS.

## Results

### Patients

At data cutoff (October 2017), data were available for 201 patients; of these, 13 patients did not have PD-L1 expression data for TCs or ICs and were excluded from the analysis. PD-L1 expression data were available for 188 patients. The percentage of patients with PD-L1 TC, IC, and TC/IC expression that exceeded the predefined cutoffs for high expression decreased at higher cutoffs (Fig 1).

### Survival

OS was based on 109 events (54% maturity), with median follow-up time in censored patients of 15.8 months. Observed data suggested that higher PD-L1 expression was associated with longer OS. Kaplan–Meier curves for OS showed greater separation between PD-L1–high and–low expressing patients at different cutoffs for ICs compared with TCs, and with combined TC/IC versus either TCs or ICs (Figs 2–4). The largest separation in the OS curves occurred for the combined TC/IC algorithm at cutoffs of $\geq 25\%/\geq 25\%$ and $\geq 50\%/\geq 25\%$. IC PD-L1 expression was associated with better median OS compared with TC PD-L1 expression at any cutoff (Fig 5). Consistent with Kaplan–Meier OS data, the difference in median OS between PD-L1–high and–low expressing patients appeared to be optimal for ICs and TC/IC at cutoffs of $\geq 25\%$ and $\geq 50\%$ (IC), and $\geq 25\%/\geq 25\%$ and $\geq 50\%/\geq 25\%$ (TC/IC) (Fig 5).

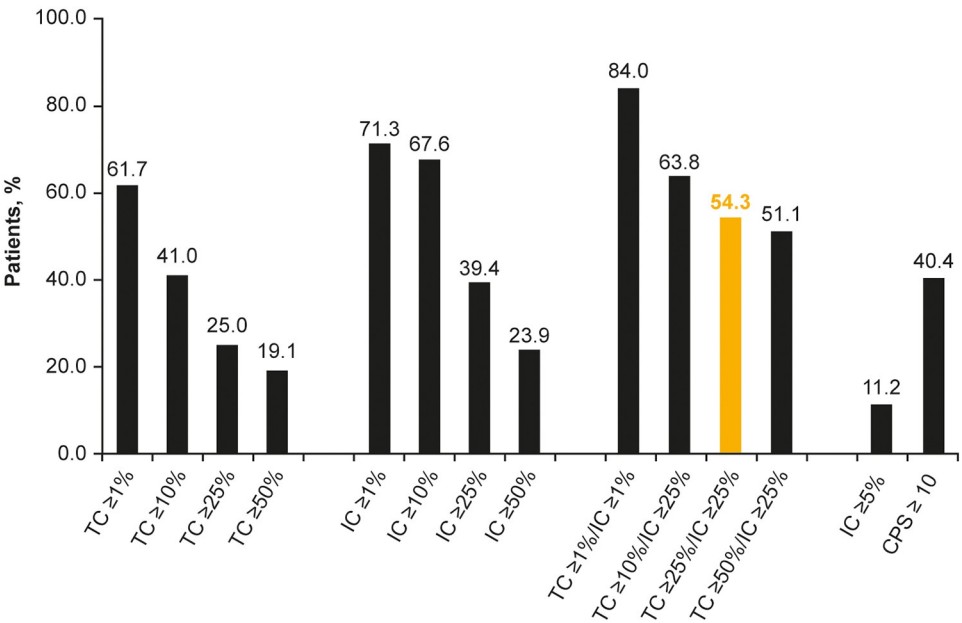

**Fig 1. Prevalence of PD-L1–high patients at different TC and IC cutoffs for defining positive PD-L1 expression in study 1108.** IC, tumor-infiltrating immune cells; PD-L1, programmed death ligand-1; TC, tumor cells. Highlighted bar indicates the PD-L1 cutoff previously used to investigate response to durvalumab.

Concordance index (c-index), revealed that for the TCs, $\geq$ 25% had the highest c-index (0.514), and for IC, $\geq$ 25% and $\geq$ 50% were highest (0.698 and 0.711, respectively) (S2 Table).

PFS was based on 163 events (81% maturity), with median follow-up time in censored patients of 14.6 months. As for OS, the observed data suggested that higher PD-L1 expression was associated with longer PFS, with greater separation between PD-L1–high and–low expressing patients at the different IC cutoffs compared with TC cutoffs, and with combined TC/IC versus TC (S1 Fig).

## Objective response rate

Thirty-five patients had an objective response (ORR, 17.4% [35/201]). Of the 188 PD-L1 expression available patients, the ORR was 17.6% (33/188). At cutoffs of $\geq$ 25% and $\geq$ 50%, IC PD-L1 expression was associated with higher ORR compared with TC PD-L1 expression (Fig 6). IC PD-L1 expression was also associated with greater separation in ORR between PD-L1–high and–low expressing patients compared with TC PD-L1 expression. This separation was also observed for the combined TC/IC algorithms.

## Positive and negative predictive values

Cutoffs at IC $\geq$ 25%, TC $\geq$ 25/IC $\geq$ 25%, and IC $\geq$ 50% were associated with the highest positive predictive values (PPV; 28%, 28%, and 50%, respectively). The highest negative predictive values (NPV), were seen with IC $\geq$ 1% (95%), TC $\geq$ 1%/IC $\geq$ 1% (100%), and TC $\geq$ 25%/ IC $\geq$ 25% (94%). Optimizing for both NPV and PPV, TC $\geq$ 25%/IC $\geq$ 25% provided the optimal cutoff (Table 1).

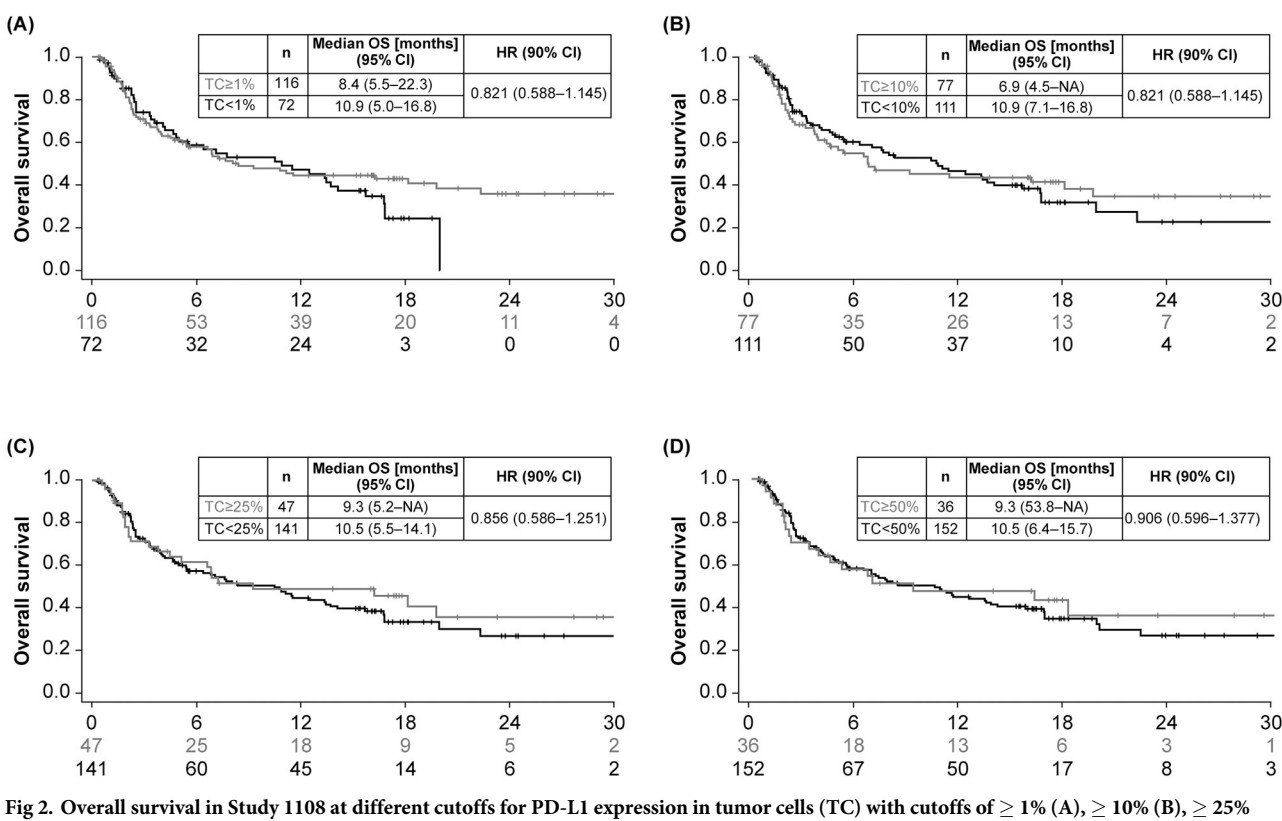

**Fig 2. Overall survival in Study 1108 at different cutoffs for PD-L1 expression in tumor cells (TC) with cutoffs of ≥ 1% (A), ≥ 10% (B), ≥ 25% (C), and ≥ 50% (D).** CI, confidence interval; HR, hazard ratio; NA, not applicable; OS, overall survival; PD-L1, programmed death ligand-1; TC, tumor cell.

## Correlation between PD-L1 expression and therapeutic outcome

Regression analysis was conducted to explore the potential association between ICs, TCs, and combined IC/TC PD-L1 expression on efficacy outcomes of durvalumab monotherapy. IC PD-L1 expression was significantly associated in univariate analyses with OS (Wald $P < 0.001$), PFS ($P < 0.001$), ORR ($P < 0.001$), best tumor size percentage decrease ($P = 0.007$), and tumor size shrinkage response at a ≥ 20% cutoff ($P = 0.006$) (Table 2). These outcome measures all remained statistically significant ($P \leq 0.05$) in multivariate analysis. In contrast, TC PD-L1 expression was only significantly associated with best tumor size percentage decrease ($P = 0.02$) in multivariate analysis, although a similar directionality of benefit was seen as for IC PD-L1 in other outcome measurements.

Evaluation of the interaction between TCs and ICs similarly found no significant association with OS, PFS, ORR, best tumor size percentage decrease, or ≥ 20% tumor size shrinkage response, indicating a lack of interaction and consequently independent effects of TCs and ICs in the prediction of efficacy. Additionally, significant association ($P \leq 0.04$) between IC PD-L1 expression and all outcome measures was observed when covariates of Bellmunt score and Memorial Sloan-Kettering Cancer Center score were included in the model.

## Optimal predictive algorithm TC and IC algorithm

Based on greater separation in the OS and PFS curves, longer median OS and higher ORR in patients with PD-L1–high expression, the best outcomes for OS, PFS, and ORR were obtained

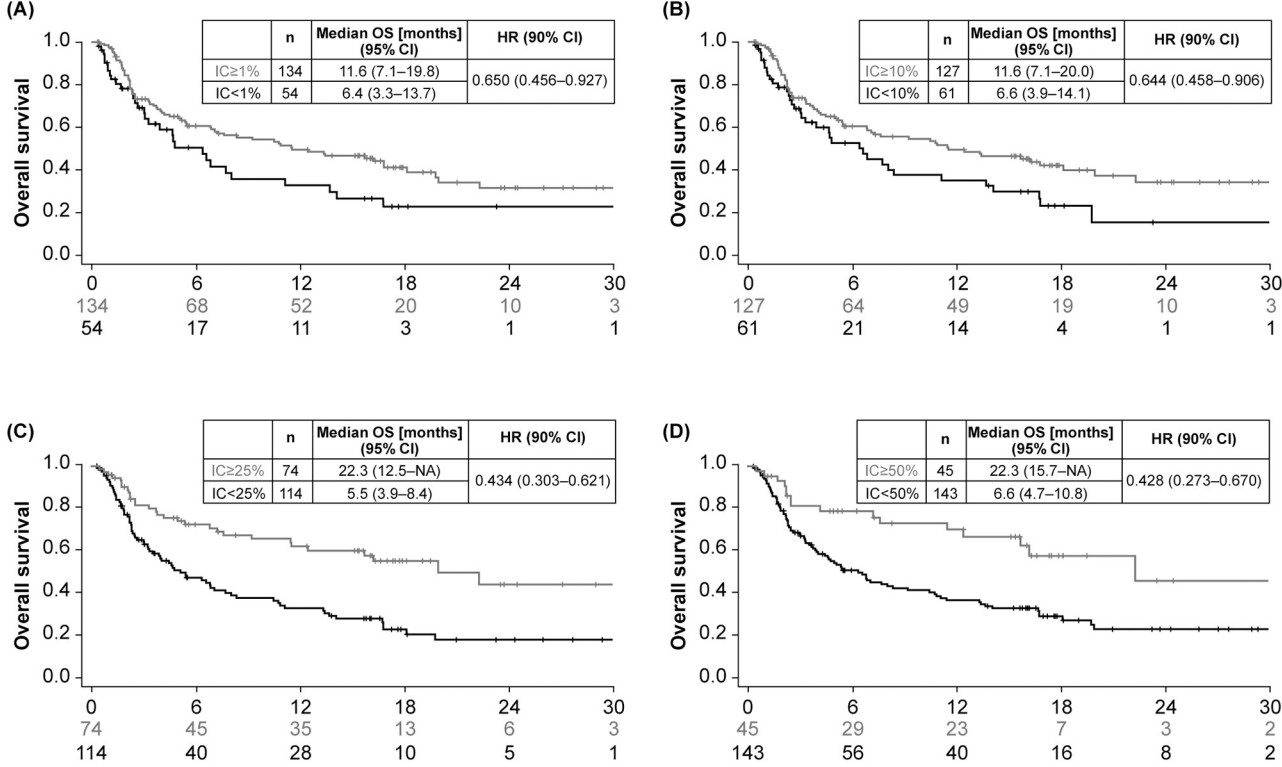

**Fig 3. Overall survival in Study 1108 at different cutoffs for PD-L1 expression in tumor-infiltrating immune cells (IC) with cutoffs of ≥ 1% (A), ≥ 10% (B), ≥ 25% (C), and ≥ 50% (D).** CI, confidence interval; IC, tumor-infiltrating immune cell; HR, hazard ratio; NA, not applicable; OS, overall survival; PD-L1, programmed death ligand-1.

when the TC and IC algorithms were combined. The TC ≥ 25%/IC ≥ 25% algorithm proved to be optimal when taking into account sensitivity and specificity, with median OS in PD-L1–high and–low patients of 19.8 versus 4.8 months, respectively (hazard ratio, 0.460; 95% confidence interval, 0.330–0.639), and ORR of 27.5% versus 5.8%.

### Findings for CPS ≥ 10 and IC ≥ 5% cutoffs

Based on IC ≥ 5% and CPS ≥ 10 cutoffs, 11.2% and 40.4% of patients, respectively, had PD-L1–high expression (Fig 1). As for other cutoffs, OS was longer for samples from PD-L1–high patients compared with PD-L1–low patients categorized using both the CPS ≥ 10 and especially the IC ≥ 5% cutoffs (Fig 3; S2 Fig). For the CPS ≥ 10 cutoff, ORR was 25% for patients categorized as PD-L1–high compared with 13% for patients with PD-L1–low expression. For the IC ≥ 5% cutoff, ORR was 48% for patients categorized as PD-L1–high compared with 14% for patients with PD-L1–low expression (Fig 4).

### Discussion

The FDA and EMA have recently cautioned against the use of single-agent checkpoint inhibition for the treatment of PD-L1–low UC following preliminary data showing decreased survival with pembrolizumab or atezolizumab compared with chemotherapy as first-line therapy in patients with PD-L1–low expression (CPS < 10 and IC < 5%, respectively) [11]. The clinical utility of PD-L1 expression is thus established in the first-line treatment of metastatic UC in cisplatin-ineligible patients according to FDA/EMA restrictions. This reinforces the need to

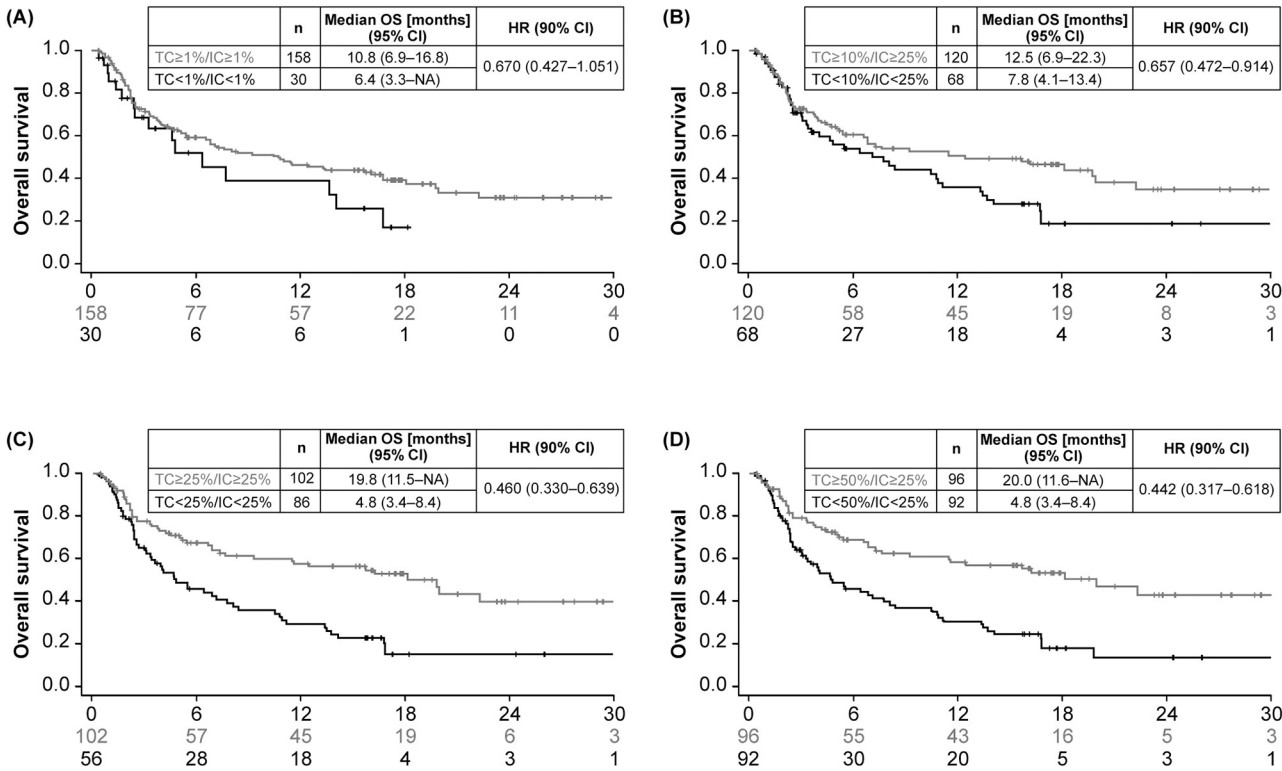

**Fig 4. Overall survival in Study 1108 at different cutoffs for PD-L1 expression in tumor cells (TC) or tumor-infiltrating immune cells (IC) with cutoffs of ≥ 1%/≥ 1% (A), ≥ 10%/≥ 25% (B), ≥ 25%/≥ 25% (C), and ≥ 50%/≥ 25% (D).** CI, confidence interval; IC, tumor-infiltrating immune cell; HR, hazard ratio; NA, not applicable; OS, overall survival; PD-L1, programmed death ligand-1; TC, tumor cell.

accurately determine PD-L1 expression based on an optimized algorithm for TCs, ICs, or their combination and using an appropriate diagnostic assay. This has been achieved in the present analysis by retrospectively applying different algorithms for PD-L1 expression to the VENTANA (SP263) Assay and using efficacy data from a single-arm trial of durvalumab monotherapy in the second- and subsequent-line post-platinum setting. Thus, the present findings are applicable to durvalumab and care should be taken in extrapolating these findings to other IO agents for which different PD-L1 assays are used. However, the comparison reinforces TC ≥ 25%/IC ≥ 25% as the optimal algorithm for the VENTANA SP263 Assay in predicting response to durvalumab monotherapy and optimizing patient care.

In the present study, we show that both IC and TC PD-L1 expression are important in predicting treatment response in patients with locally advanced/metastatic UC, as assessed using the VENTANA (SP263) Assay. While in multivariate analysis IC but not TC PD-L1 expression was significantly associated with OS, PFS, and ORR ($P < 0.001$ for each), interaction analysis showed a similar directionality of benefit for ICs and TCs. The lack of interaction between the effects of PD-L1 expression on TCs and ICs suggests that both cell populations need to be considered in any algorithm used to define PD-L1 expression in UC. Indeed, confirming previous reports [7,25], the use of the combined TC ≥ 25%/IC ≥ 25% algorithm appears to provide optimal predictive value. The selection of this algorithm was based on higher median OS and ORR in patients with PD-L1–high expression at this TC/IC cutoff, on separation of the OS Kaplan–Meier curves and differences in ORR among patients with PD-L1–high compared with PD-L1–low TC/IC staining as well as prevalence and optimal PPV/NPV values. OS and

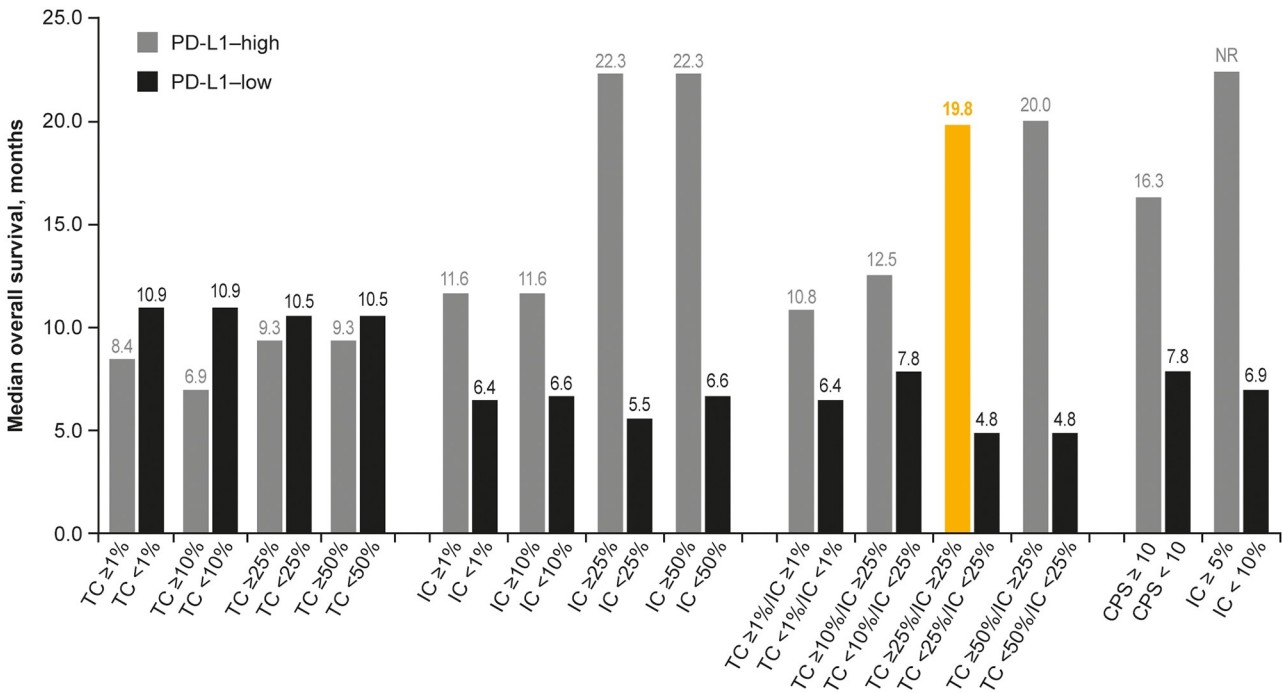

**Fig 5. Median overall survival in Study 1108 at different TC and IC cutoffs for defining positive (PD-L1–high) and negative (PD-L1–low) expression.** IC, tumor-infiltrating immune cells; PD-L1, programmed death ligand-1; TC, tumor cells. Highlighted bar indicates the PD-L1 cutoff previously used to investigate response to durvalumab.

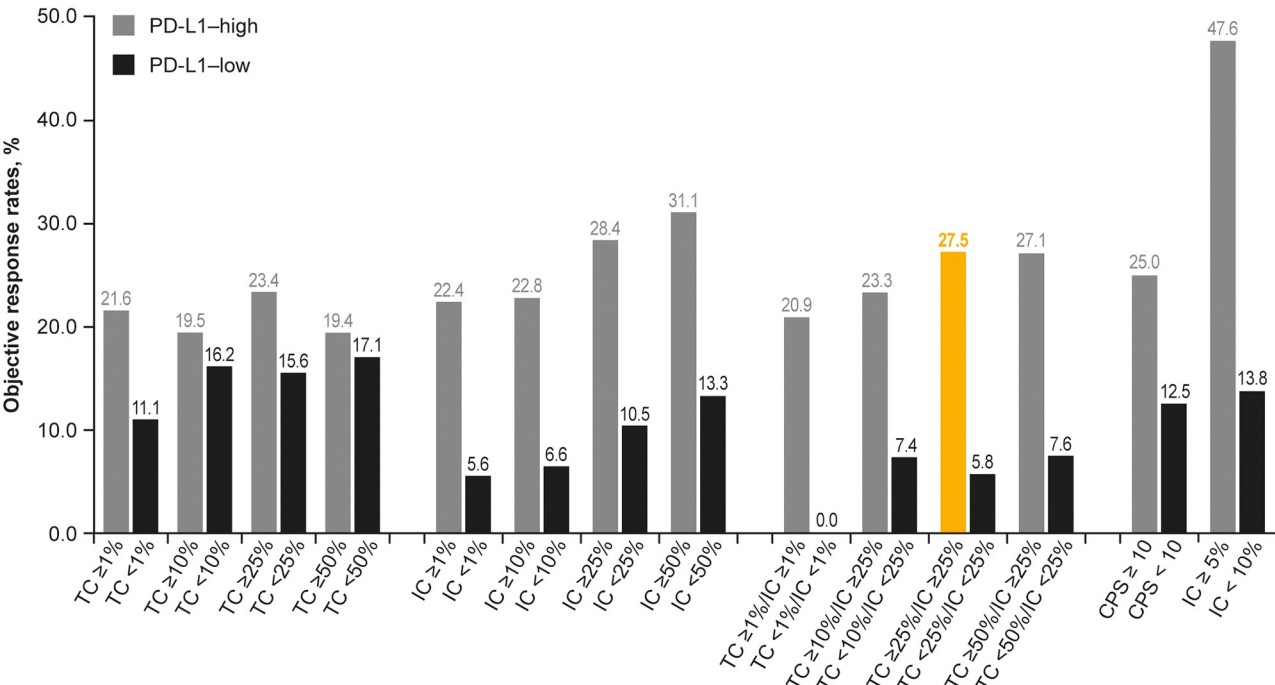

**Fig 6. Objective response rates in study 1108 at different TC and IC cutoffs for defining positive (PD-L1–high) and negative (PD-L1–low) expression.** IC, tumor-infiltrating immune cells; PD-L1, programmed death ligand-1; TC, tumor cells. Highlighted bar indicates the PD-L1 cutoff previously used to investigate response to durvalumab.

**Table 1. Positive and negative predictive value of responders and non-responders in PD-L1 high and low negative patients.**

| Cutoff % | PD-L1 high responders n | PD-L1 high Nonresponders n | Positive predictive value (95% CI) | PD-L1 low/negative Responders n | PD-L1 low/negative Nonresponders n | Negative predictive value (95% CI) |
|---|---|---|---|---|---|---|
| TC ≥ 1% | 25 | 91 | 0.22 (0.14–0.29) | 8 | 64 | 0.89 (0.82–0.96) |
| TC ≥ 10% | 15 | 62 | 0.20 (0.11–0.28) | 18 | 93 | 0.84 (0.77–0.91) |
| TC ≥ 25% | 11 | 36 | 0.23 (0.11–0.36) | 22 | 119 | 0.84 (0.78–0.90) |
| TC ≥ 50% | 7 | 29 | 0.19 (0.07–0.32) | 26 | 126 | 0.83 (0.77–0.89) |
| IC ≥ 1% | 30 | 104 | 0.22 (0.15–0.29) | 3 | 51 | 0.94 (0.88–0.01) |
| IC ≥ 10% | 29 | 98 | 0.23 (0.16–0.30) | 4 | 57 | 0.93 (0.87–1.00) |
| IC ≥ 25% | 21 | 53 | 0.28 (0.18–0.39) | 12 | 102 | 0.90 (0.84–0.95) |
| IC ≥ 50% | 14 | 31 | 0.31 (0.18–0.45) | 19 | 124 | 0.87 (0.81–0.92) |
| TC ≥ 1%/ IC ≥ 1% | 33 | 125 | 0.21 (0.15–0.27) | 0 | 30 | 1 (1) |
| TC ≥ 10%/ IC ≥ 25% | 28 | 92 | 0.23 (0.16–0.31) | 5 | 63 | 0.93 (0.86–0.99) |
| TC ≥ 25%/ IC ≥ 25% | 28 | 74 | 0.28 (0.19–0.36) | 5 | 81 | 0.94 (0.89–0.99) |
| TC ≥ 50%/ IC ≥ 25% | 26 | 70 | 0.27 (0.18–0.36) | 7 | 85 | 0.92 (0.87–0.98) |

CI, confidence interval; IC, tumor-infiltrating immune cell; TC, tumor cell.

PFS was also prolonged for PD-L1–high compared with PD-L1–low patients when samples were categorized using CPS ≥ 10 and IC ≥ 5%. Of particular note, the TC ≥ 25%/IC ≥ 25% cutoff appeared to be optimal for separating responders from nonresponders in the PD-L1–low population. This contrasts with the IC ≥ 5% cutoff, which although enriching for response in patients with PD-L1–high expression, included a substantial (14%) proportion of responders with PD-L1–low expression. CPS ≥ 10 proved to be a less effective predictor of response

**Table 2. PD-L1 biomarker regression and interaction analyses in the 1108 study.**

| Model | | Overall survival | | Progression-free survival | | Objective response rate | | Best tumor size % change from baseline | | Tumor size shrinkage response (20% shrinkage in tumor cutoff) | |
|---|---|---|---|---|---|---|---|---|---|---|---|
| | | HR | Wald P | HR | Wald P | Logit | Wald P | Linear | Wald P | Logit | Wald P |
| **Regression analysis** | | | | | | | | | | | |
| Univariate TC | | 0.999 | 0.63 | 0.999 | 0.60 | 0.003 | 0.55 | –0.198 | 0.12 | 0.006 | 0.22 |
| Multivariate TC[a] | | 0.995 | 0.13 | 0.995 | 0.08 | 0.006 | 0.36 | –0.274 | 0.02 | 0.011 | 0.07 |
| Univariate IC | | 0.986 | < 0.001 | 0.989 | < 0.001 | 0.022 | < 0.001 | –0.404 | 0.007 | 0.017 | 0.006 |
| Multivariate IC[a] | | 0.989 | 0.02 | 0.991 | 0.01 | 0.021 | 0.01 | –0.300 | 0.03 | 0.014 | 0.05 |
| **Interaction analysis** | | | | | | | | | | | |
| TC IC TC*IC | TC | 0.999 | 0.86 | 0.996 | 0.3 | 0.016 | 0.06 | –0.237 | 0.19 | 0.010 | 0.19 |
| | IC | 0.987 | 0.005 | 0.988 | 0.001 | 0.030 | < 0.001 | –0.431 | 0.01 | 0.019 | 0.008 |
| | TC*IC | 1.000 | 0.68 | 1.000 | 0.55 | 0.000 | 0.12 | 0.0005 | 0.93 | –0.0001 | 0.71 |
| TC IC TC*IC + covariates[a] | TC | 0.997 | 0.55 | 0.992 | 0.05 | 0.026 | 0.01 | –0.372 | 0.02 | 0.018 | 0.05 |
| | IC | 0.991 | 0.07 | 0.989 | 0.005 | 0.034 | 0.001 | –0.309 | 0.04 | 0.019 | 0.03 |
| | TC*IC | 1.000 | 0.49 | 1.000 | 0.40 | –0.001 | 0.03 | 0.003 | 0.54 | –0.0002 | 0.42 |

[a]Other covariates: Bellmunt score, MSKCC score.

IC, tumor-infiltrating immune cell; HR, hazard ratio; MSKCC, Memorial Sloan-Kettering Cancer Center; TC, tumor cell; TC*IC, combined TC and IC.

and survival with durvalumab when applied to the VENTANA SP263 Assay. The nature of the CPS ≥ 10 algorithm means it is more sensitive to low levels of TC PD-L1 expression than IC expression. For example, a patient with no IC PD-L1 expression, but ≥ 10% TC PD-L1 expression will be classified as high by only this algorithm. Given that TC PD-L1 expression showed a weaker impact on outcomes compared with IC PD-L1 expression, these patients may account for the CPS ≥ 10 algorithm being less effective in predicting response.

As for durvalumab, which is partnered with the VENTANA (SP263) Assay, the diagnostic anti–PD-L1 monoclonal antibodies SP142 and 22C3 have been partnered with atezolizumab (anti–PD-L1) and pembrolizumab (anti–PD-1), respectively, to evaluate the association between PD-L1 expression and OS in patients with advanced/metastatic UC. In patients who had progressed after first-line platinum-based chemotherapy and who were subsequently treated with atezolizumab monotherapy, ORR was 15% and median OS 7.9 months. Similar to the present analysis, IC PD-L1–high expression enriched for responders, with an ORR of 26% and median OS of 11 months [28]. Also in the second- and subsequent-line setting, the median OS among patients treated with pembrolizumab was 10.3 months compared with 7.4 months for chemotherapy; among those with TC PD-L1–high expression, median OS was 8.0 months compared with 5.2 months for pembrolizumab and chemotherapy, respectively [18]. Although median OS was superior for pembrolizumab compared with chemotherapy irrespective of PD-L1 expression, it is interesting that the cutoff used for PD-L1–high expression (CPS ≥ 10) did not result in improved median OS compared with the overall pembrolizumab-treated population. Treatment of patients with refractory metastatic UC using another therapeutic anti–PD-L1 monoclonal antibody, avelumab, also failed to provide an OS benefit for PD-L1–high patients (TC ≥ 5% using the 73–10 anti–PD-L1 clone) compared with PD-L1–low patients [6]. However, in a single-arm phase 2 study investigating first-line pembrolizumab, long-term follow-up data showed prolonged median OS of 18.5 months for patients with PD-L1–high expression based on the CPS ≥ 10 cutoff compared with 11.5 months for all patients [29].

A limitation of our study is that it was a single arm study, and therefore we cannot discount that PD-L1–high expression may have had confounding effects in terms of prolonged survival. Data regarding the potential impact of PD-L1 expression on survival in UC are conflicting. Krabbe *et al* 2017 showed that TC PD-L1 expression in a subset of resected UC patients was negatively associated with survival [30].

In contrast, Bellmunt *et al* 2015 showed that PD-L1 expression on IC but not TC was associated with better survival in patients with metastatic UC treated with platinum chemotherapy [31]. Based on the literature, both TC and IC PD-L1 expression may have effects on survival in UC implying that caution is needed with respect to the use of PD-L1 expression to predict outcomes to treatment [18,31–34]. Optimally, the value of PD-L1 as a biomarker would be determined in a randomized controlled trial where it is possible to evaluate survival outcomes in relation to treatment and also in patient subgroups.

A further limitation of this study is that its findings may not be generalizable to other IO agents and PD-L1 diagnostic assays. We attempted to address this by applying other leading algorithms (CPS ≥ 10 and IC ≥ 5%) to our study. It should be noted that these algorithms were applied to slides stained with the VENTANA SP263 Assay and derived from captured raw expression data, rather than scored directly. Although published data suggests that the SP263 Assay is analytically similar to 22C3, the SP142 Assay has shown differences in sensitivity, particularly for TC PD-L1 expression. Therefore, patients defined as PD-L1–high using the IC ≥ 5% in our study may differ from those who would have been classed as PD-L1–high in studies using the VENTANA SP142 Assay. However, in our study, the TC ≥ 25%/IC ≥ 25% cutoff was better than all other algorithms at predicting response to durvalumab when applied to a single assay (the VENTANA SP263 Assay).

In conclusion, these findings suggest that both IC and TC PD-L1 expression may have predictive value for a range of outcomes in patients with locally advanced/metastatic UC, with a lack of interaction suggesting that the value of IC PD-L1 expression is independent of TC PD-L1 expression. Although the assessment of IC PD-L1 expression appears to have a higher association with better efficacy outcomes in UC compared with TC PD-L1 expression, the combination of TC/IC and in particular the selected cutoff of TC $\geq$ 25%/IC $\geq$ 25% may optimally be used in conjunction with the VENTANA (SP263) Assay to select patients who are most likely to respond to durvalumab treatment in the second-line setting. These findings continue to support the selection of the TC $\geq$ 25%/IC $\geq$ 25% cut off. Due to the differences among assays these findings may not be directly applicable to other IO agents. Therefore, further data particularly from large, randomized controlled trials are needed to confirm an optimal algorithm with predictive validity in UC.

## Supporting information

**S1 Table. Full name of the ethics committee/institutional review board(s) that approved the study.**
(DOCX)

**S2 Table. Concordance index for overall survival based on cutoffs for PD-L1 expression.**
(DOCX)

**S1 Fig. Progression-free survival in Study 1108 based on illustrative cutoffs for PD-L1 expression.** Tumor cells $\geq$ 25% (TC $\geq$ 25%; A), tumor-infiltrating immune cells $\geq$ 25% (IC $\geq$ 25%; B), and TC $\geq$ 25% or IC $\geq$ 25% (C).
(TIF)

**S2 Fig. Overall survival in Study 1108 based on cutoffs for PD-L1 expression of CPS $\geq$ 10 (A) and IC $\geq$ 5% (B).** CPS $\geq$ 10, combined positive score defined as the percentage of PD-L1–expressing TC and IC relative to the total number of TC, with CPS $\geq$ 10 considered to be PD-L1–high; IC, tumor-infiltrating immune cells; IC $\geq$ 5%, $\geq$ 5% PD-L1 expression on IC; TC, tumor cells.
(TIF)

## Acknowledgments

The authors would like to thank the patients, their families and caregivers, and all investigators involved in the studies. The authors would also like to thank Marianne Ratcliffe (AstraZeneca) for her input. Medical writing support, which was in accordance with Good Publication Practice (GPP3) guidelines, was provided by Sherri Baber of Parexel (Hackensack, NJ).

## Author Contributions

**Conceptualization:** Magdalena Zajac, Pralay Mukhopadhyay, J. Andrew Williams, Jill Walker.

**Data curation:** Yong Ben, Joyce Antal, Ashok K. Gupta.

**Formal analysis:** Jiabu Ye, Xiaoping Jin.

**Investigation:** Xiaoping Jin, Yong Ben, Joyce Antal, Ashok K. Gupta, Marlon C. Rebelatto.

**Project administration:** Ashok K. Gupta, Jill Walker.

**Supervision:** Pralay Mukhopadhyay, Jill Walker.

**Validation:** Jiabu Ye, Pralay Mukhopadhyay, Xiaoping Jin.

**Visualization:** Jiabu Ye.

**Writing – original draft:** Magdalena Zajac, Jiabu Ye, J. Andrew Williams, Jill Walker.

**Writing – review & editing:** Magdalena Zajac, Jiabu Ye, Pralay Mukhopadhyay, Xiaoping Jin, Yong Ben, Joyce Antal, Ashok K. Gupta, Marlon C. Rebelatto, J. Andrew Williams, Jill Walker.

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
