## [Decision Letter · Decision Letter 0]

18 Feb 2020

PONE-D-19-18286

Optimal PD-L1–high cutoff for association with overall survival in patients with urothelial cancer treated with durvalumab monotherapy

PLOS ONE

Dear Mrs Packham,

Thank you for submitting your manuscript to PLOS ONE. After careful consideration, we feel that it has merit but does not fully meet PLOS ONE’s publication criteria as it currently stands. Therefore, we invite you to submit a revised version of the manuscript that addresses the points raised during the review process.

We would appreciate receiving your revised manuscript by Apr 03 2020 11:59PM. To enhance the reproducibility of your results, we recommend that if applicable you deposit your laboratory protocols in protocols.io, where a protocol can be assigned its own identifier (DOI) such that it can be cited independently in the future. For instructions see: http://journals.plos.org/plosone/s/submission-guidelines#loc-laboratory-protocols

We look forward to receiving your revised manuscript.

Kind regards,

Hyunseok Kang, MD, MPH

Academic Editor

PLOS ONE

Additional Editor Comments (if provided):

I am so sorry about the delay in the review process. Many reviewers have declined the review for conflict of interest.

Journal Requirements:

3. Thank you for providing the following Funding Statement: 

"This study was funded by AstraZeneca. Medical writing support, which was in accordance with Good Publication Practice (GPP3) guidelines, was provided by Sherri Baber of PAREXEL (Hackensack, NJ) and was funded by AstraZeneca. "

We note that one or more of the authors is affiliated with the funding organization, indicating the funder may have had some role in the design, data collection, analysis or preparation of your manuscript for publication; in other words, the funder played an indirect role through the participation of the co-authors.

If the funding organization did not play a role in the study design, data collection and analysis, decision to publish, or preparation of the manuscript and only provided financial support in the form of authors' salaries and/or research materials, please review your statements relating to the author contributions, and ensure you have specifically and accurately indicated the role(s) that these authors had in your study in the Author Contributions section of the online submission form. Please make any necessary amendments directly within this section of the online submission form.  Please also update your Funding Statement to include the following statement: “The funder provided support in the form of salaries for authors [insert relevant initials], but did not have any additional role in the study design, data collection and analysis, decision to publish, or preparation of the manuscript. The specific roles of these authors are articulated in the ‘author contributions’ section.”

If the funding organization did have an additional role, please state and explain that role within your Funding Statement.

Please also provide an updated Competing Interests Statement declaring this commercial affiliation along with any other relevant declarations relating to employment, consultancy, patents, products in development, or marketed products, etc.  

4. Please amend the manuscript submission data (via Edit Submission) to include author Jill Walker.

5. Please amend your authorship list in your manuscript file to include author Lana Packham.

6. Thank you for stating the following in the Competing Interests section:

"M. Zajac, J. Ye, P. Mukhopadhyay, Y. Ben, and J. Walker are employees of AstraZeneca. X. Jin, J. Antal, A. K. Gupta, M. C. Rebelatto, and J. A.Williams are employees of MedImmune. "

We note that one or more of the authors are employed by a commercial company: AstraZeneca and MedImmune.

7. We note that you have included the phrase “data not shown” in your manuscript. Unfortunately, this does not meet our data sharing requirements. PLOS does not permit references to inaccessible data. We require that authors provide all relevant data within the paper, Supporting Information files, or in an acceptable, public repository. Please add a citation to support this phrase or upload the data that corresponds with these findings to a stable repository (such as Figshare or Dryad) and provide and URLs, DOIs, or accession numbers that may be used to access these data. Or, if the data are not a core part of the research being presented in your study, we ask that you remove the phrase that refers to these data.

8. Please upload a copy of Figure 5-6, to which you refer in your text on page 10-11. If the figure is no longer to be included as part of the submission please remove all reference to it within the text.

Reviewers' comments:

Reviewer's Responses to Questions

**Comments to the Author**

1. Is the manuscript technically sound, and do the data support the conclusions?

Reviewer #1: Yes

Reviewer #2: Yes

2. Has the statistical analysis been performed appropriately and rigorously? 

Reviewer #1: I Don't Know

Reviewer #2: Yes

3. Have the authors made all data underlying the findings in their manuscript fully available?

Reviewer #1: Yes

Reviewer #2: Yes

4. Is the manuscript presented in an intelligible fashion and written in standard English?

Reviewer #1: Yes

Reviewer #2: Yes

5. Review Comments to the Author

Reviewer #1: Overall well written. Robust survival data set.

Labeling of the Figures should be re-done for consistency with the text. Especially Figure 2 and its sub-letters.

Line 242: Objective Response Rate and this section.

Authors should present the data in the following tablet format, to calculate PPV and NPV. The bar graphs in Figure 6 are inadequate when critiquing its role as a predictive biomarker of response.

A good predictive biomarker has both positive predictive value and negative predictive value.

Responders n =33 Non- responders n = 155

PD-L1 High (cutoff __)n=__ ## #

PD-L1 Low (cuffoff __)n=__ # ###

Line 290 and 298: “ IC> 5% cut-off..” does not appear to be consistent with the labeling of the Figures 1 and 6. Confirm that “CPS 10” and “IC2/3” on the Figures refer to CPS >10 and IC> 5%, respectively. If so, update the figure.

Line 334: Consider rephrasing this sentence: “TC>25%/IC>25% cutoff appeared to be optimal for excluding responders from the PD-L1-low population.” This cutoff separates the responders from non-responders.

Reviewer #2: Zajac et al reported the manuscript entitled as “Optimal PD-L1–high cutoff for association with overall survival in patients with 2 urothelial cancer treated with durvalumab monotherapy”. The authors performed post hoc analysis of the single-arm, phase 1/2 Study 1108 (NCT01693562), using using the VENTANA (SP263) IHC, and tumor samples obtained prior to durvalumab. The primary objective was to determine whether the TC≥ 25%/IC≥ 25% algorithm (cutoff of 25% TC or 25% IC with PD-L1 staining at any intensity above background) was optimal for predicting response to durvalumab. The authors analyzed the predictive value of PD-L1 with various cutoff level.

This manuscript contains important issue regarding optimal cutoff level of PD-L1 expression. The manuscript is well written and scientifically sound. I’d like give one comment.

1. C-statistics and C- index

Basically, the authors find the optimal cutoff for the largest separation in the OS curves occurred. Not only minimize P-value and but also maximize the separation of OS curve, various cutoff values were categorized relative arbitrary. In order to overcome arbitrary cutoff, C statistics can be used. C-index for each cutoff level could measure the separability of OS curve likely with AUROC. It had better measure C-index

6. PLOS authors have the option to publish the peer review history of their article (what does this mean?). If published, this will include your full peer review and any attached files.

Reviewer #1: No

Reviewer #2: No

---

## [Author Response · Author response to Decision Letter 0]

30 Mar 2020

Please refer to the uploaded Response to Reviewers document in the Attach Files section.

---

## [Editor Report · Decision Letter 1]

6 Apr 2020

Optimal PD-L1–high cutoff for association with overall survival in patients with urothelial cancer treated with durvalumab monotherapy

PONE-D-19-18286R1

Dear Dr. Walker,

We are pleased to inform you that your manuscript has been judged scientifically suitable for publication and will be formally accepted for publication once it complies with all outstanding technical requirements.

With kind regards,

Hyunseok Kang, MD, MPH

Academic Editor

PLOS ONE

Additional Editor Comments (optional):

Thank you for addressing all the comments.
---

## [Editor Report · Acceptance letter]

13 Apr 2020

PONE-D-19-18286R1 

Optimal PD-L1–high cutoff for association with overall survival in patients with urothelial cancer treated with durvalumab monotherapy 

Dear Dr. Walker:

I am pleased to inform you that your manuscript has been deemed suitable for publication in PLOS ONE. Congratulations! Your manuscript is now with our production department. 

With kind regards,

on behalf of

Dr. Hyunseok Kang 

Academic Editor

PLOS ONE